# A Multi-Bit Quantization Low-Latency Voltage Sense Amplifier Applied in RRAM Computing-in-Memory Macro Circuits

Wei Hu [ID], Hangze Zhang, Rongshan Wei [ID] and Qunchao Chen *

School of Physics and Information Engineering, Fuzhou University, Fuzhou 350116, China;
whu@fzu.edu.cn (W.H.); 211127053@fzu.edu.cn (H.Z.); wrs08@fzu.edu.cn (R.W.)
* Correspondence: qcchen@fzu.edu.cn

**Abstract:** Conventional sense amplifiers limit the performance of current RRAM computing-in-memory (CIM) macro circuits, resulting in high latency and energy consumption. This paper introduces a multi-bit quantization technology low-latency voltage sense amplifier (MQL-VSA). Firstly, the multi-bit quantization technology enhances circuit quantization efficiency, reducing the number of operational states in conventional VSA. Secondly, by simplifying the sequential logic circuits in conventional VSA, the complexity of sequential control signals is reduced, further diminishing readout latency. Experimental results demonstrate that the MQL-VSA achieves a 1.40-times decrease in readout latency and a 1.28-times reduction in power consumption compared to conventional VSA. Additionally, an 8-bit input, 8-bit weight, 14-bit output macro circuit utilizing MQL-VSA exhibited a 1.11times latency reduction and 1.04-times energy savings.

**Keywords:** RRAM; computing-in-memory; voltage sense amplifier; low latency

## 1. Introduction

In recent years, AI chips based on computing-in-memory (CIM) architecture are increasingly being integrated into cloud computing, the Internet of Things (IoT), and AI edge devices, demonstrating revolutionary advantages. Among various CIM architectures, those based on resistive random access memory (RRAM) arrays have garnered widespread attention in both academia and industry due to their high integration, low power consumption, low readout latency, and compatibility with traditional CMOS processes [1–3]. This makes RRAM-based CIM architectures a potent area for research and application. Nonetheless, the majority of CIM architectures employ traditional ADCs for quantization, where the high resolution of these ADCs leads to greater latency and energy consumption, thus limiting further performance improvements [4–6].

Consequently, an increasing number of researchers are adopting sense amplifiers (SA) as quantizers, replacing traditional ADCs. Feraj Husain proposed a parallel-sensing multi-level SA based on a 65nm CMOS process [7]. Wang Ye et al. developed a reference-subtracting current sense amplifier that not only saves power but also achieves higher linearity and a smaller common-mode input range. However, due to its serial data readout scheme, there is margin for improvement in the readout latency of this CSA [8]. Byung-Kwon An designed a CSA based on dynamic reference, which showed significant improvements in power and latency performance compared to conventional CSA [9]. Hua Zhang et al. implemented a bit-line-clamping circuit to enhance the pre-charging speed of VSA circuits, reducing readout latency at the expense of additional power consumption and area [10].

The research above indicates that while SAs offer significant power advantages over traditional ADCs, they are limited by low resolution, quantizing only one bit per cycle. This single-bit quantization necessitates multiple cycles for multi-bit digital code quantization, leading to redundancy and complexity in the circuit's operational states, sequential control

signals, and decoding circuits. Consequently, conventional SA readout latency still accounts for 40% of the total operational delay in CIM architectures [11]. Thus, reducing SA readout latency is crucial for enhancing the speed of CIM macro circuits.

To address the high latency issue of SAs, we propose a multi-bit quantization technology low-latency voltage sense amplifier (MQL-VSA):

(1)　The introduction of multi-bit quantization technology allows for two-bit digital output in a single quantization cycle using two voltage references, improving upon the inefficiency of conventional VSA's one-bit-per-cycle mode. This reduces the number of operational states and thereby decreases readout latency;

(2)　The use of simple combinational logic circuits for two-bit quantization avoids the complexity of comparators, registers, and other circuits found in conventional VSA, simplifying the design and reducing the complexity of sequential control signals, further optimizing readout latency performance.

## 2. Architecture of CIM Macro Based on RRAM Arrays

### 2.1. Composition of the Architecture

In current research, CIM architectures based on RRAM arrays predominantly consist of two components, as depicted in Figure 1 [12]. The CIM architecture is primarily composed of an RRAM array (Weight Array in Figure 1) and peripheral circuits (Peripheral Circuit in Figure 1), where:

(1)　The RRAM array stores the convolution kernel's weight information (W[k:0]) and reads the stored weights via the WL drive module;

(2)　The peripheral circuit is responsible for performing subsequent multiply-accumulate (MAC) operations. Under the control of sequential signals generated by the CIM Control module, the analog output from the RRAM array is quantized by the VSA circuit, eventually outputting the digital result of multiple multiplication calculations ($\sum_{i=0}^{n} IN[m:0] \times W$). Thus, the readout latency of the VSA significantly impacts the system's overall performance.

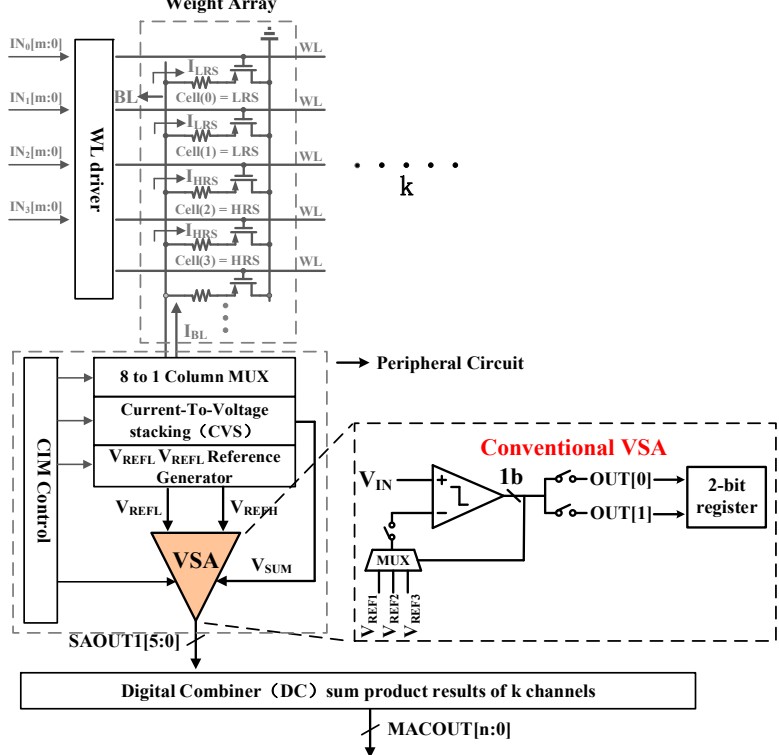

**Figure 1.** CIM macro architecture based on RRAM array.

The digital combiner (DC) performs the final weighted summation (MACOUT[n:0]) of the results from k channels, facilitating complete matrix convolution operations.

### 2.2. Architectural Computation Sequence

Combining with Figures 1 and 2, the computational process in CIM architectures initiates when the system clock CLK arrives, activating the RRAM array. RRAM array interacts with the external input IN[m:0], generating the $I_{BL}$ current. The $I_{BL}$ current is then fed into the CVS module (Current-to-Voltage-Stacking, CVS), which converts it into a corresponding voltage, $V_{SUM}$. Subsequently, $V_{SUM}$ is quantized in the VSA module and produces the corresponding digital code value.

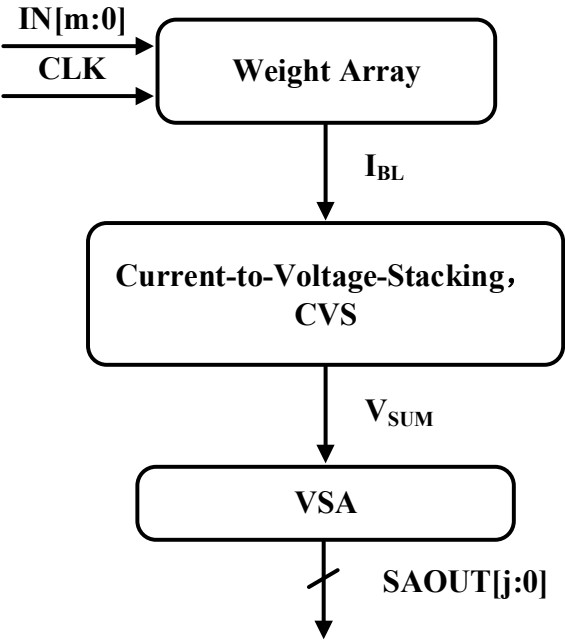

**Figure 2.** Architecture computational flowchart.

As analyzed, the essence of RRAM array-based CIM architectures lies in transforming matrix convolution operations into analog voltage multiply–accumulate operations, ultimately quantifying the analog voltage ($V_{SUM}$) into digital output using VSA. Therefore, the readout latency of VSA is critical to the speed performance of the CIM architecture.

Conventional VSAs, with their simplistic structure, quantize only one bit per cycle, leading to inefficiency and elevated readout latency, especially when dealing with high bit outputs. As illustrated in Figure 1, conventional VSA comprises comparators, multiplexers, and output registers [13]. Each quantization cycle in the VSA involves comparing $V_{IN}$ with $V_{REF}$ to produce a one-bit result, which is then stored in a register. The subsequent cycle's $V_{REF}$ is adjusted based on this output, and the comparison with $V_{IN}$ continues until all digital codes are quantified.

Although this single-bit quantization method is straightforward, it is not efficient, with conventional VSA readout latency accounts for approximately 40% of the total system cycle [11]. Hence, optimizing the readout latency of VSA could significantly enhance the system's operational speed.

### 3. The Proposed MQL-VSA

As deduced from Section 2.2, the primary reason for the high readout latency in conventional VSA circuits is the excessive number of operational states. To address this, we introduce a multi-bit quantization technology that accomplishes two-bit digital code quantization within a single cycle, thereby reducing the number of operational states in the VSA circuit.

The proposed MQL-VSA, as shown in Figure 3, comprises four main components:

(1) The sampling structure, consisting of four sampling switches (SW3–SW6) and two sampling capacitors (C0, C1), samples the voltage values of $V_{REFL}$, $V_{REFH}$, and $V_{SUM}$ ($V_{REFL}$ = 1/4 VDD, $V_{REFH}$ = 3/4 VDD, $V_{SUM}$ is the input voltage that is quantized);

(2) The LSB sensing includes three MOS transistors NO (N1), P0 (P1), P2 (P3), a switch SW1 (SW2), and an inverter, generating the OUT2 (OUT2B) signal for the LSB-detecting circuit;

(3) The latch comprises two cascaded inverters formed by N2, P4 and N3, P5 along with two switching MOS transistors (N4, P6), producing SAOUT[1] by comparing the voltages at nodes Q1B and Q1;

(4) The LSB-detecting circuit, a 2-to-1 selector, selects between OUT2 and not (OUT2B) based on the value of SAOUT[1] to determine the SAOUT[0] result.

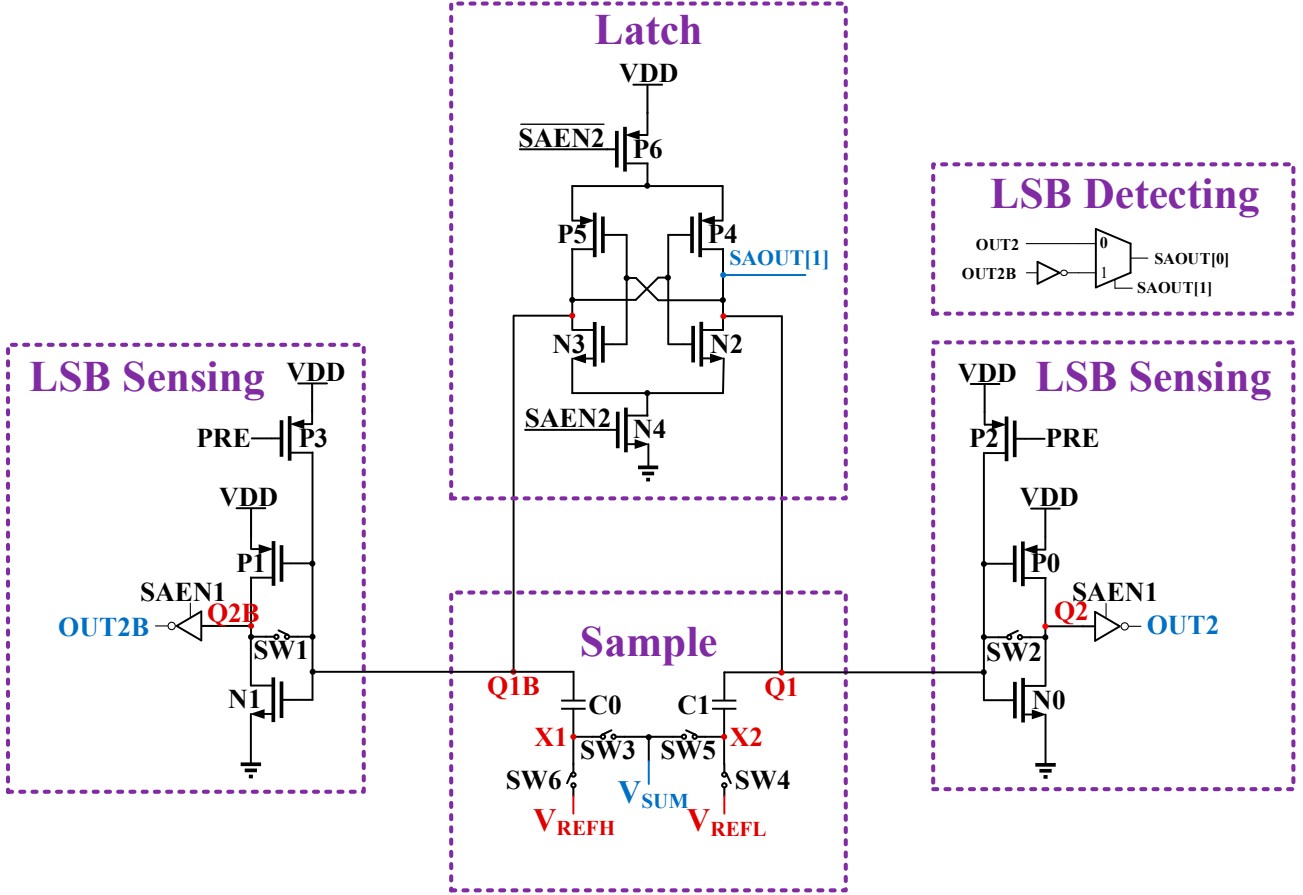

**Figure 3.** The proposed MQL-VSA circuit.

### 3.1. Architectural Computation Sequence

As deduced in Section 2.2, conventional VSAs compare a single reference voltage $V_{REF}$ with the input voltage $V_{IN}$ in each quantization cycle, yielding a one-bit output. This implies that with an increase in VSA output bits, the reference voltage $V_{REF}$ requires frequent switching, and the registers must continually store the results of each quantization cycle, leading to a multitude of operational states in the VSA circuit [14].

To address this issue, we propose a multi-bit quantization improvement. The MQL-VSA introduces two reference voltages, $V_{REFL}$ and $V_{REFH}$, where $V_{REFL}$ = 1/4 VDD and $V_{REFH}$ = 3/4 VDD. As shown in Figure 4a, $V_{REFL}$ and $V_{REFH}$ divide the quantization range into four regions, each corresponding to a specific two-bit digital code. Capacitors C0

and C1, under the control of sampling switches (SW3–SW6), sample the voltages of $V_{REFL}$, $V_{REFH}$, and $V_{SUM}$. The voltage differences at nodes X2 andX1 is given by:

$$\Delta V_1 = V_{SUM} - V_{REFL} \tag{1}$$

$$\Delta V_2 = V_{REFH} - V_{SUM} \tag{2}$$

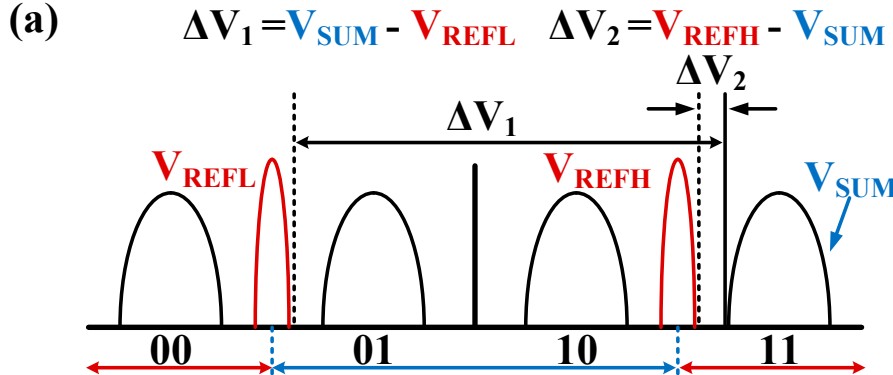

**(b)** **VSA critical status summary table**

| $V_{SUM}$ case | 00 | 01 | 10 | 11 |
|---|---|---|---|---|
| $\Delta V_1$, $\Delta V_2$ | < 0, > 0 | > 0, > 0 | > 0, > 0 | > 0, < 0 |
| $\Delta V_{Q1}$, $\Delta V_{Q1B}$ | < 0, > 0 | > 0, > 0 | > 0, > 0 | > 0, < 0 |
| $\Delta V_{Q2}$, $\Delta V_{Q2B}$ | > 0, < 0 | < 0, < 0 | < 0, < 0 | < 0, > 0 |
| SAOUT [1:0] | 00 | 01 | 10 | 11 |

**Figure 4.** (**a**) Encoding representation of MQL-VSA quantized outputs; (**b**) Tabulated Overview of Crucial Signal-Output Correlations.

As indicated by Equations (1) and (2), the sign of $\Delta V_1$ represents the magnitude relationship between $V_{SUM}$ and $V_{REFL}$, and similarly, the sign of $\Delta V_2$ indicates the relationship between $V_{SUM}$ and $V_{REFH}$. The correspondence between the signs of $\Delta V_1$ and $\Delta V_2$ and the digital codes is listed in Figure 4b. Finally, capacitors C1and C0couple $\Delta V_1$ and $\Delta V_2$, respectively, to the inputs of the latch structure. The latch, by determining the relationship between $\Delta V_1$ and $\Delta V_2$, identifies the range in which the $V_{SUM}$ falls and outputs the corresponding result.

Thus, the MQL-VSA, by introducing two reference voltages as comparison benchmarks, achieves two-bit digital code output in a single quantization cycle, reducing the operational states associated with reference voltage switching and intermediate data storage in conventional VSA, thereby shortening readout latency.

### 3.2. Workflow of the MQL-VSA

The workflow of the MQL-VSA, as depicted in Figure 5, begins with the circuit in its initial state, where all switches in Figure 3 are off and the MOS transistors are in the cut-off region. As the MQL-VSA prepares to transition to the working state, it enters the standby phase. P2 and P3 are on (PRE = 0) to pre-charge the voltages at nodes Q1B and Q1, respectively, to VDD. Subsequently, the circuit transitions to the working state, comprising three phases:

(1)　PH1. This phase carries out the sampling of $V_{SUM}$, $V_{REFL}$, and $V_{REFH}$. As shown in Figure 6a, SW3 and SW4 are on to pass $V_{SUM}$ to X1 and $V_{REFL}$ to X2, respectively. Concurrently, SW1 and SW2 are on, distributing the charge from nodes Q1B, Q1 to nodes Q2B, Q2, resulting in all four nodes attaining a voltage of 1/2 VDD;

(2)　PH2. This phase achieves voltage differencing and ΔV coupling. Illustrated in Figure 6b, SW5 and SW6 are on, resulting in a voltage swing of $\Delta V_1$ at X2and of $\Delta V_2$ atX1. Since the voltages across capacitors C0 and C1 cannot change abruptly, $\Delta V_{Q1B} = \Delta V_2$, $\Delta V_{Q1} = \Delta V_1$. Concurrently, SW1 and SW2 are turned off, and N0, P0 and N1, P1 formed inverter structures, causing voltage swings at nodes Q2B and Q2 opposite to those at Q1B and Q1, respectively;

(3)　PH3. This phase accomplishes the output of SAOUT[0] and SAOUT[1]. As Figure 6c illustrates, with SAEN1 = 1, the voltages at Q2 and Q2B are processed through inverters to generate OUT2 and OUT2B, respectively. Then, with SAEN2 = 1, the latch is activated to compare the voltages at nodes Q1andQ1B, determining the relationships of $\Delta V_1$ and $\Delta V_2$ and outputs the MSB of SAOUT[1:0]. Simultaneously, the 2-to-1 selector activates, outputting SAOUT[0] as the $\overline{OUT2B}$ when SAOUT[1] = 1 and as the OUT2 when SAOUT[1] = 0.

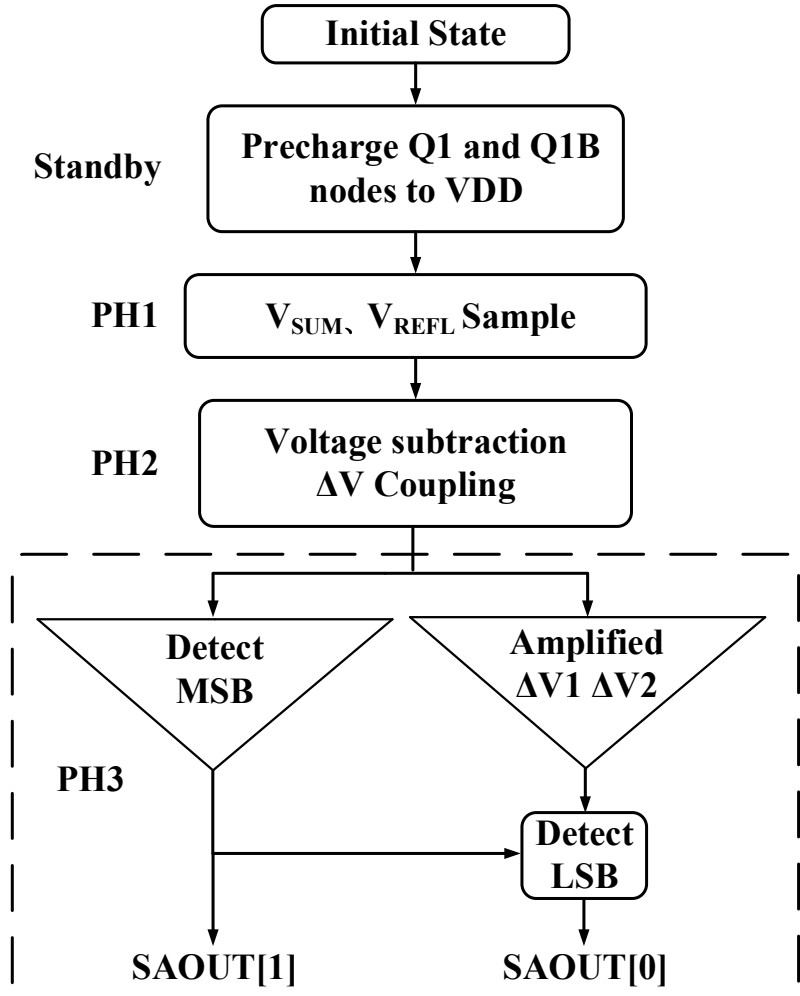

**Figure 5.** Workflow diagram of the MQL-VSA.

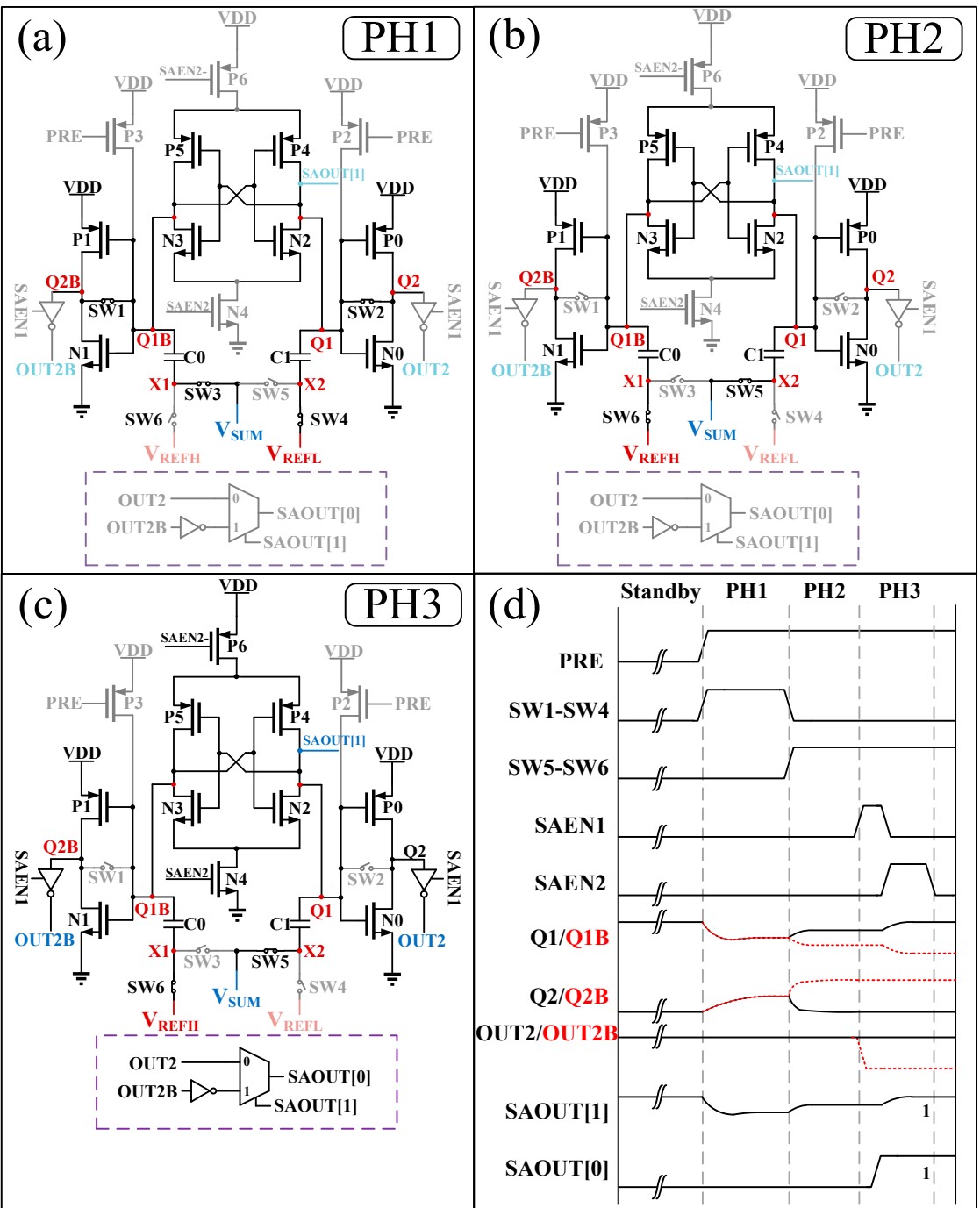

**Figure 6.** Sequential schematic of MQL-VSA circuit: (**a**) PH1; (**b**) PH2; (**c**) PH3; (**d**) control signal sequence for MQL-VSA output quantized as "11".

Thus, MQL-VSA completes a two-bit quantization cycle and then resets to the initial state, preparing for the next quantization cycle.

### 3.3. Sizing Methodology for MOS Transistors, Switches, and Capacitors in MQL-VSA

Sections 3.1 and 3.2 discussed the circuit architecture and operational principles of MQL-VSA. This section will detail the method for determining the dimensions of MOS transistors, switches, and capacitors within the circuit. It primarily conducts a theoretical analysis of the latch, LSB-sensing, and sampling modules in the circuit depicted in Figure 3. The specifics are as follows:

(1) Latch module. The latch module in Figure 3 consists of inverters sequentially connected, formed by N3, P5 and N2, P4. Its equivalent circuit is illustrated in Figure 7. Due to the circuit's symmetrical structure, the transconductance ($g_m$), output impedance ($R_{out}$), and load capacitance ($C_L$) of the two inverters are equal. Based on Kirchhoff's Law and the analysis method for first-order circuit time domain responses [15], we can derive the following at nodes X and Y:

$$g_m V_x + \frac{V_y}{R_{out}} + C_L \frac{dV_y}{dt} = 0 \tag{3}$$

$$g_m V_y + \frac{V_x}{R_{out}} + C_L \frac{dV_x}{dt} = 0 \tag{4}$$

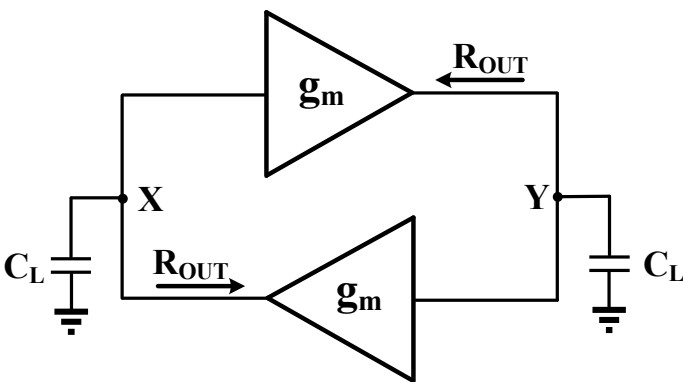

**Figure 7.** Equivalent circuit of the latch module.

Defining $A = g_m R_{out}$, $\tau = R_{out} C_L$, the voltage difference $\Delta V$ between nodes X and Y can be determined as:

$$\Delta V = V_x - V_y = \frac{\tau}{A-1} \frac{d\Delta V}{dt} \approx \frac{\tau}{A} \frac{d\Delta V}{dt} \tag{5}$$

Based on Equation (5), the expression for $\Delta V$ is determined as:

$$\Delta V = \Delta V_0 \times e^{T/\tau} \tag{6}$$

In Equation (6), $\Delta V_0$ represents the initial voltage difference between points X and Y. When the operating period of the MQL-VSA is determined, the required time constraint "T" can be established. Thus, by adjusting the W/L of the MOS transistors in the latch module, the value of $\tau$ at nodes X and Y can be controlled, ensuring the latch module completes a comparison within the time "T".

When the W/L of the MOS transistors increases, the MOS transistor's on-resistance has a more significant impact on $\tau$ than the parasitic capacitance [16]. Therefore, enlarging the MOS transistor size appropriately can reduce the comparison delay of the latch module, meeting the setup time requirements.

(2) LSB-sensing module. As shown in Figure 3, the primary function of this module is to sense the voltage changes at node Q1 (Q1B). Thus, during the PH1 phase of the circuit, the trip-point voltage $V_{TRIP}$ of the inverter made up of N1, P1 (N0, P0) should equal the $V_{Q1B}$ voltage at that time, which is $V_{TRIP} = V_{Q1B} = 1/2$ VDD. Due to the different mobilities of N-channel and P-channel MOSFETs, the W/L of P1 (P0) should be about 4 to 5 times that of N1 (N0) to set the inverter's trip-point voltage $V_{TRIP}$ to 1/2 VDD. P2 (P3) is used as a charging transistor, so the smallest size is adequate.

(3) Sampling module. According to Figure 3, when the capacitors and switches are in operation, they can be equivalently analyzed as a low-pass filter. The equivalent

circuit is shown in Figure 8, where $R_{on}$ represents the on-resistance of the switch. Since the switches in Figure 3 are all transmission gate structures, $R_{on}$ can be expressed as:

$$R_{on} = R_{on,PMOS} \parallel R_{on,NMOS} \tag{7}$$

$$R_{on,NMOS} = \frac{1}{\mu_n C_{ox}(W/L)_N(VDD - V_{IN} - V_{THN})} \tag{8}$$

$$R_{on,PMOS} = \frac{1}{\mu_p C_{ox}(W/L)_P(V_{IN} - |V_{THP}|)} \tag{9}$$

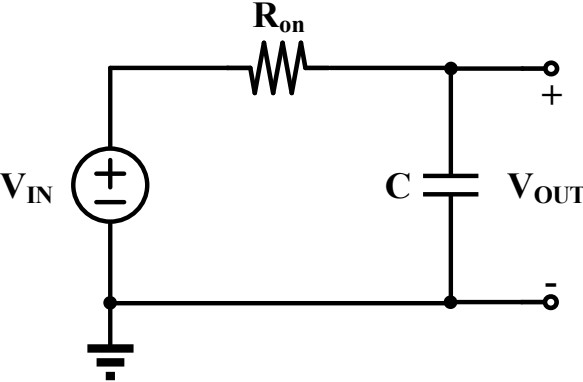

**Figure 8.** Equivalent circuit of the low-pass filter.

During the voltage build-up process, defining the $V_{OUT}$ voltage reaching 99% of its intended target as the steady state, the constraint relationship can be expressed as:

$$R_{on} \times C < \frac{T_S}{2\ln 10} \tag{10}$$

According to Equation (10), to meet the setup time requirements, it is necessary to increase the size of the transmission gates to reduce $R_{on}$. The choice of capacitance size involves a trade-off. Under the premise of satisfying setup time constraints, the capacitance should be increased appropriately so that it significantly exceeds the parasitic capacitance at nodes X1 (X2), Q1 (Q1B), aiming to minimize error as much as possible.

We have now finished the theoretical analysis of the latch, LSB-sensing, and sampling modules in Figure 3. Based on Equations (3)–(10) and subsequent practical calculations, the dimensions of all MOS transistors, switches, and capacitors have been summarized in Table 1.

**Table 1.** Summary of MOS transistor/switch/capacitor sizes.

| MOS Transistor/Switch/Capacitor | Size |
| --- | --- |
| N0, N1 | W = 220 nm, L = 180 nm |
| N2–N4 | W = 880 nm, L = 180 nm |
| P0, P1 | W = 970 nm, L = 180 nm |
| P2, P3 | W = 220 nm, L = 180 nm |
| P4–P6 | W = 880 nm, L = 180 nm |
| SW1–SW6 | PMOS: W = 7.5 μm, L = 180 nm<br>NMOS: W = 2.5 μm, L = 180 nm |
| C0 C1 | W = 10 μm, L = 10 μm<br>Capacitance = 197.5 fF |

### 3.4. Advantages of the MQL-VSA in Readout Latency

As demonstrated in Figure 9, for the quantization of two-bit data, the proposed MQL-VSA, compared to conventional VSA, eliminates intermediate states associated with reference voltage switching and intermediate data storage (as shown in PH2, PH3, and PH5 in Figure 9b), this reduction lowers the number of operational states from six to three. In conventional VSA, the process of switching and stabilizing the reference voltage, often constrained by the analog buffer and circuit parasitic, results in considerable latency, significantly impeding the readout latency performance of conventional VSA [17].

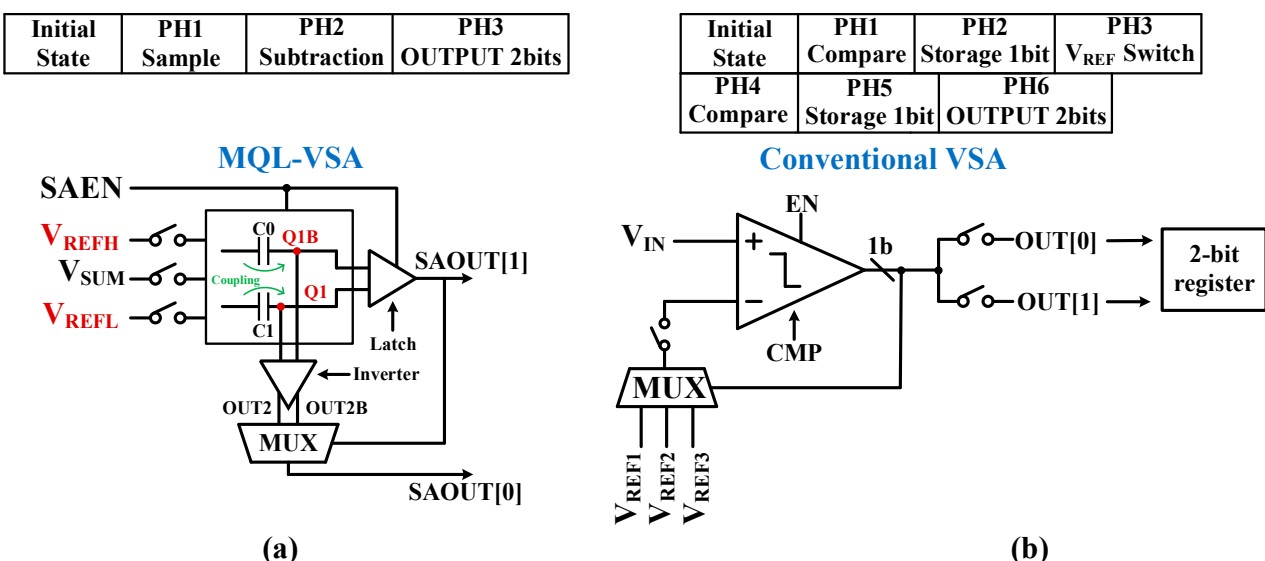

**Figure 9.** Comparison of operational states in MQL-VSA and conventional VSA for 2-bit quantization: (**a**) structure of MQL-VSA; (**b**) structure of conventional VSA.

The MQL-VSA, by incorporating multi-bit quantization technology, reduces the delay associated with three operational states, particularly avoiding the high latency induced by reference voltage switching. Furthermore, its parallel quantization of two-bit data is indisputably more efficient than the serial operations employed by conventional VSA. Additionally, the MQL-VSA omits the use of multi-bit registers (two-bit register) and multiplexers (MUX) or other sequential logic circuits, thereby circumventing the additional delays caused by complex gate circuits. It also obviates the need for consideration of inter-module timing coordination, which leads to waiting times, thus further optimizing the circuit's latency. Consequently, MQL-VSA demonstrates superior performance in terms of readout latency.

## 4. Experiments Result and Comparison

### 4.1. Verification of MQL-VSA Functionality

This section conducts experimental validation of the MQL-VSA circuit using the MXIC L18B 180 nm CMOS process, with a supply voltage of VDD = 1.8 V. The layout of the CIM macro circuit and the designed MQL-VSA are shown in Figure 10, with the CIM macro circuit and MQL-VSA total size being 2630 μm × 2149 μm and 170 μm × 135 μm, respectively. The validation encompasses two main aspects:

(1) Testing the functionality and readout latency performance of the MQL-VSA;
(2) Evaluating the readout latency characteristics of the RRAM array-based CIM macro circuit which utilizes MQL-VSA. The macro circuit employs the CIM architecture shown in Figure 10, with parameters set to $m = 8$, $k = 8$, and $n = 14$, forming an 8-bit input, 8-bit weight, 14-bit output CIM macro. The RRAM array consists of $TiN/HfO_2/TiN$-based binary memristors [18,19], with the memristor unit simulated using the MuHAM model [20–22]. The array size is 0.5-Mb (1024 rows × 512 columns).

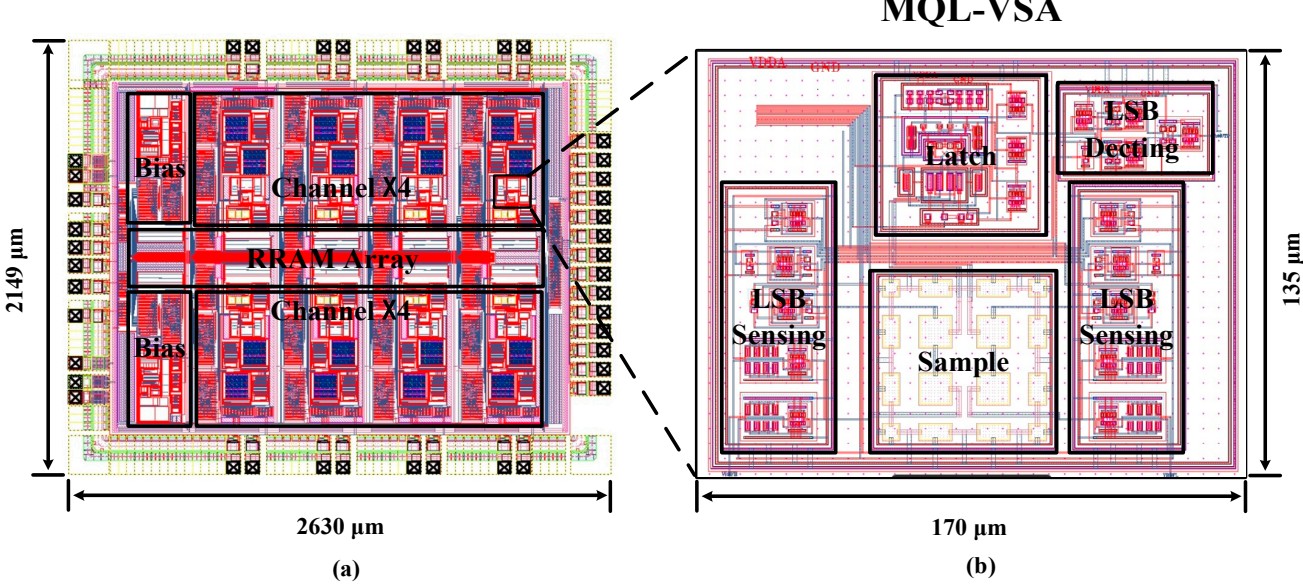

**Figure 10.** The layout of CIM macro circuit and the designed MQL-VSA: (**a**) the layout of CIM macro circuit; (**b**) the layout of MQL-VSA.

Based on the experimental results from (1) and (2), we will expand a comparison between conventional VSA and MQL-VSA in terms of readout latency and power consumption.

The functional verification of the MQL-VSA is illustrated in Figure 11. The specific signals function as follows: SAOUT[0] and SAOUT[1] represent the MSB and the LSB of the 2-bit output, respectively; PRE, SW12, SW56, SW78, SAEN1, and SAEN2 serve as the control signals of the circuit; the voltage signals Q1 (Q1B) are crucial for outputting SAOUT[1]; Q2 (Q2B) and OUT2 (OUT2B) act as inputs for the LSB Detecting, generating SAOUT[0] (details in Section 3.2).

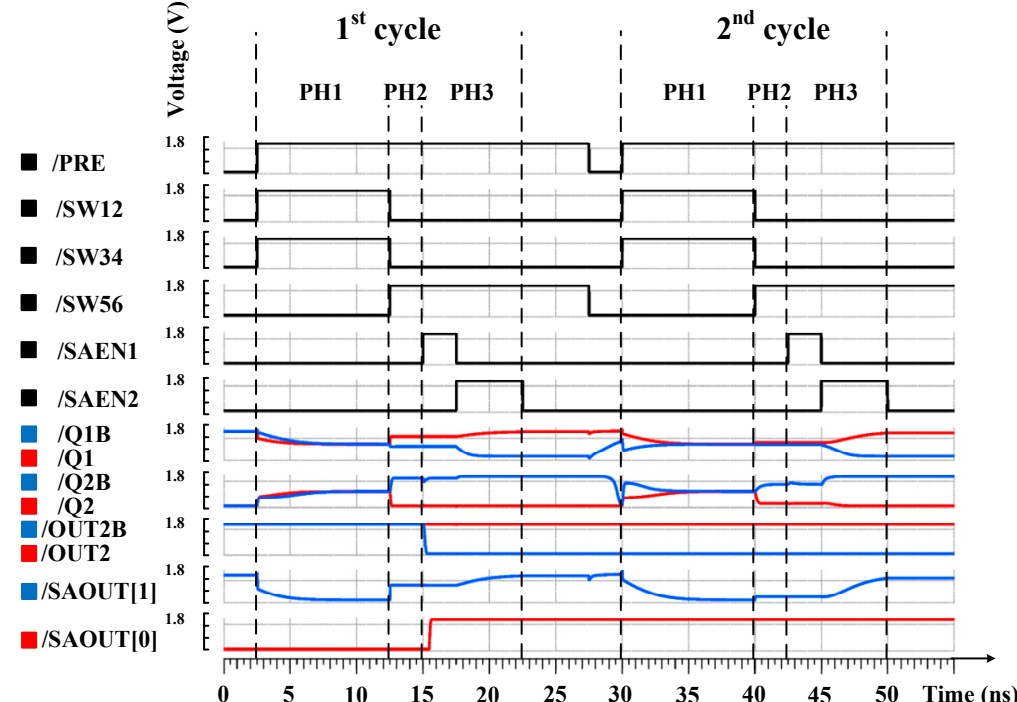

**Figure 11.** Voltage–time domain waveforms of crucial signals when $V_{SUM}$ = 1.7 V is quantized as 1111 by MQL-VSA.

When $V_{SUM}$ = 1.7 V, the quantization process of MQL-VSA is as follows:

(1)    In the first cycle, with VREFH = 1.35 V and VREFL = 0.45 V, the quantization results of MQL-VSA are SAOUT[1]=1 and SAOUT[0] = 1, respectively;

(2)    In the second cycle, with VREFH = 1.6875 V and VREFL = 1.4625 V, the quantization results are SAOUT[1] = 1 and SAOUT[0] = 1.

The voltage time-domain waveforms of the crucial signals in Figure 11 demonstrate that the MQL-VSA accurately achieves the quantization function while reducing readout latency, meeting the anticipated requirements.

Furthermore, to mitigate the impact of overshoots during switch transitions on the circuit's normal operation, this experiment implemented measures to minimize these effects. Specifically:

(1)    For the digital signals in Figure 11 (OUT2, OUT2B, SAOUT[0]), all digital signals in the experiment pass through a digital buffer to filter overshoots before output, resulting in clean and well-defined signals in Figure 11.

(2)    To reduce overshoots in the analog signals of Figure 11 (including Q1, Q1B, Q2, Q2B, SAOUT[1]), a non-overlapping control signal for all switches was employed. For example, during the transition from PH1 to PH2 phase, the falling edge of SW12 and SW34 and the rising edge of SW56 are non-overlapping. This prevents simultaneous conduction of switches during state transitions, thereby only minor overshoots occur in the analog signals, which are clear and well-defined, and these overshoots do not affect the final quantization outcome.

Thus, by addressing signal overshoots, the MQL-VSA demonstrates significant resilience against such non-ideal factors, which stands as one of its major advantages.

Significantly, this study conducted PVT and 200 Monte Carlo tests on the MQL-VSA across a broad spectrum of input voltages. This involved step scanning in 10 mV increments within a 0–1.8 V range, yielding 180 voltage inputs. The findings of these experiments reveal:

(1)    In the PVT experiments, five MOS corners (ff, tt, ss, sf, fs), three capacitor corners (ff, tt, ss), three power supply voltages (1.44 V, 1.8 V, and 2.16 V, which represent a ±20% fluctuation in supply voltage), and three temperature environments (−40 °C, 27 °C, 125 °C) were selected, totaling 135 scenarios. In all 135 different PVT conditions, the MQL-VSA consistently produced accurate quantization results. For instance, with input voltages of 0.36 V, 0.99 V, and 1.7 V, the MQL-VSA output the binary codes 0010, 1001, and 1111, respectively, in each of the 135 PVT experiments, demonstrating robust PVT characteristics.

(2)    In the Monte Carlo simulations, the MQL-VSA underwent 200 tests, consistently yielding correct quantization results. For example, with input voltages of 0.36 V, 0.99 V, and 1.7 V, the MQL-VSA output the binary codes 0010, 1001, and 1111, respectively, in each of the 200 tests. This indicates that even in the presence of device mismatches, the MQL-VSA maintains its accurate quantization capabilities.

As analyzed and evidenced by the experimental results, the proposed MQL-VSA, through two identical quantization cycles, accomplishes the quantization of 4-bit data. Each 2-bit output reduces the delay of three operational states compared to conventional VSA. Thus, the entire 4-bit quantization process diminishes the delay of six operational states, effectively reducing the readout latency of VSA through multi-bit quantization technology.

### 4.2. Performance Comparison of MQL-VSA and Conventional VSA

As depicted in Figure 12, when quantizing 4-bit data, the overall readout latency of the MQL-VSA circuit, in comparison to the conventional VSA [10], is reduced from 70 ns to 50 ns, a decrease by a factor of 1.40. This reduction in readout latency is attributed to the fewer operational states required by MQL-VSA, eliminating the need for frequent switching of reference voltage and intermediate data storage prevalent in conventional VSA. Moreover, the overall power consumption of the circuit is reduced from 90.42 μW to

70.64 µW, a decrease by a factor of 1.28. This reduction is due to the fewer operational states in MQL-VSA, significantly reducing the need for frequent switching within the circuit and consequently lowering dynamic power consumption.

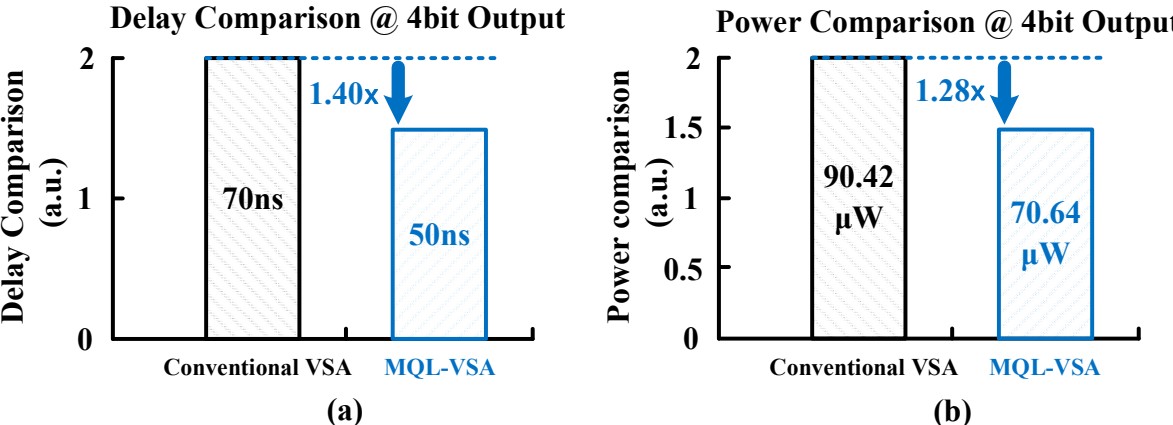

**Figure 12.** Comparison of readout delay and power consumption between MQL-VSA and conventional VSA circuit [10] (illustrated with 4-bit quantized output): (**a**) delay comparison; (**b**) power comparison.

Note that since the power consumption of the clock circuit is low, it only occupies a small part of the power consumption of the VSA and does not affect the final comparison result, so the power consumption information shown in Figure 12 does not include the clock circuit part.

### 4.3. Comparison Based on CIM Architectures

As shown in Figure 13, when comparing macro circuits utilizing conventional VSA [10] with those using MQL-VSA, the overall computational latency of the macro circuit incorporating MQL-VSA is reduced from 830 ns to 750 ns, a decrease by a factor of 1.11. Concurrently, the power consumption is lowered from 4.06 mW to 3.90 mW, a reduction by a factor of 1.04. The relatively modest decrease in macro circuit power consumption can be attributed to the RRAM array, which accounts for a significant portion of the power usage. As a result, the power consumption reduction contributed by MQL-VSA is less pronounced, leading to a less substantial overall decrease in power consumption for the macro circuit.

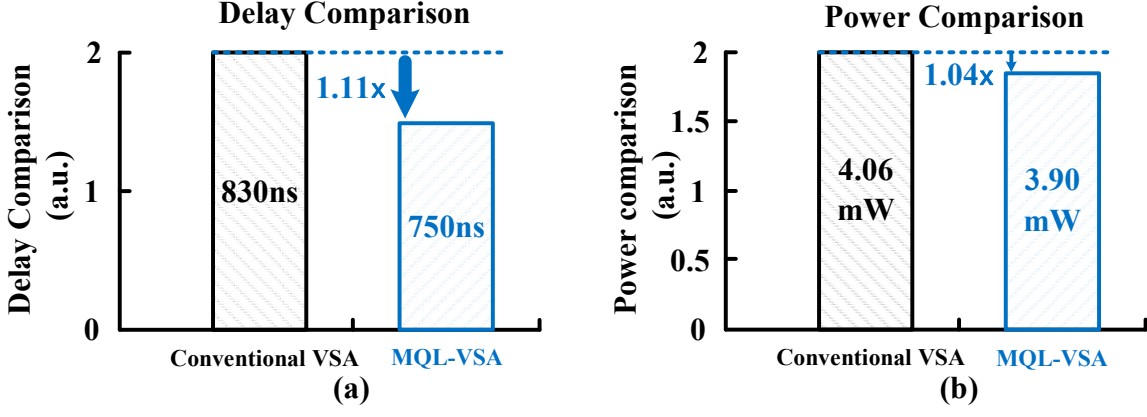

**Figure 13.** Delay–power consumption comparison between MQL-VSA and conventional VSA circuit [10] (based on an RRAM array with CIM macro for 8-bit input, 8-bit weight, and 14-bit output): (**a**) delay comparison; (**b**) power comparison.

*4.4. Comparison with Previous SAs*

Table 2 summarizes the performance of sense amplifiers (SAs) from other studies and compares them with the proposed MQL-VSA. The results indicate that in the context of quantizing 4-bit data, the MQL-VSA's performance in terms of readout latency and power consumption is comparable to other works utilizing advanced processes.

**Table 2.** Performance summary and comparison.

|  | [7] | [8] | [9] | [10] | **This Work** |
|---|---|---|---|---|---|
| Technology node (nm) | 65 | 28 | 40 | 130 | 180 |
| Memory | N/A | RRAM | RRAM | Flash | RRAM |
| Supply voltage (V) | 1 | 0.8 | 1.1 | 1.2 | 1.8 |
| Sensing approach | Current | Current | Current | Voltage | Voltage |
| Power (μW) | 59.38 | 7.132 | 48.44 | 90.42 | 70.64 |
| Readout latency (ns) | 30 | 26.4 | 14 | 70 | 50 |
| Quantify bit/cycle | 1 | 1 | 1 | 1 | 2 |
| FoM * | 3.65 | 14.87 | 5.90 | 2.05 | 10.19 |

* FoM = 100 × (Technology node) × (Quantify bit/cycle)/[(Power) × (Readout Delay)].

To provide a more comprehensive assessment of SA circuit performance, the concept of FoM is introduced in Table 2. The FoM is designed to evaluate the performance of SA circuits from multiple perspectives, including process, quantization bits, power consumption, and readout latency. As seen from Table 2, the FoM value of the proposed MQL-VSA significantly surpasses that of the studies in references [7,9,10], indicating its superior overall performance. However, the proposed MQL-VSA does have limitations when compared to the 28 nm high-speed, low-voltage process used in reference [8]. Despite this, the mature commercial 180 nm process, with its lower design complexity and cost, offers considerable advantages for the widespread adoption and application in computing-in-memory chips.

It is noteworthy that the MQL-VSA is also compatible with advanced processes below 130 nm. This compatibility stems from the use of standard devices, such as MOS transistors and MIM capacitors, which operate under large signal conditions. Unlike traditional analog circuits, the biasing requirements for MQL-VSA are less stringent, allowing it to function effectively in the low-voltage modes of advanced processes. Compared to the ultra-deep submicron processes utilized in references [7–10], the proposed MQL-VSA based on the mature commercial 180 nm process from MXIC significantly reduces chip manufacturing costs and development complexity. This makes it more conducive to the proliferation and application of computing-in-memory chips. Furthermore, as process technology advances, there is substantial margin for improvement in the readout latency and power consumption performance of MQL-VSA.

## 5. Discussion

Based on the above analysis and experimental validation, the proposed multi-bit quantization technology effectively reduces the excessive number of operational states in conventional VSA. This reduction avoids lengthy readout latency caused by frequent reference voltage switching and intermediate data storage in registers. Additionally, the MQL-VSA's simplified structure results in superior power and area consumption compared to conventional VSA.

(1) Power Consumption: as analyzed in Section 3, the MQL-VSA's simplified circuit leads to lower power consumption. Experimental results from Section 4 show that the circuit's power consumption decreased from 90.42 μW in conventional VSA to 70.64 μW in MQL-VSA, a 1.28-times reduction;

(2) Area: as seen from Figure 3, the MQL-VSA circuit uses simple components like inverters, switches, capacitors, and latch structures, avoiding the complex circuit

elements like dynamic comparators and multiplexers found in conventional VSA. This simplicity also translates to a reduction in area usage;

(3) Common mode input range: conventional VSA structures employing dynamic comparators are limited in their common-mode input level (which must exceed the threshold voltage of the comparator's input pair). Thus, they require an additional common-mode level input to function correctly. However, MQL-VSA, utilizing a latch structure instead of dynamic comparators, has no such limitation in its input common-mode range, indirectly reducing the circuit's power consumption.

Essentially, VSA is a circuit that converts analog voltage into digital signals. Thus, the proposed MQL-VSA is also applicable in other scenarios requiring analog-to-digital conversion, especially in applications demanding low readout latency and low power consumption. Furthermore, by configuring control signals, MQL-VSA can perform multi-cycle quantization to achieve higher bit outputs, allowing customization based on different application requirements.

## 6. Conclusions

This paper introduces a multi-bit quantization technology low-latency voltage sense amplifier (MQL-VSA) and validates its functionality and performance using a CIM macro circuit based on RRAM arrays. Compared to conventional VSA, the proposed MQL-VSA:

(1) Utilizes multi-bit quantization technology to reduce the number of operational states in each quantization cycle by quantizing 2-bit data within one cycle;

(2) Employs combinational logic circuits and Latch structures for MSB and LSB detection, decreasing the complexity of sequential control signals and further optimizing readout latency while simplifying the circuit structure.

Experimental results demonstrate that the MQL-VSA reduces latency by 1.40 times and power consumption by 1.28 times compared to conventional VSA. In CIM macro circuits based on RRAM arrays, using MQL-VSA reduces system latency by 1.11 times and power consumption by 1.04 times, indicating effective enhancement in latency performance for CIM circuits.

The proposed MQL-VSA not only improves readout latency but also shows improvements in area and power consumption. It is suitable for new computing-in-memory architectures and high-speed signal acquisition systems requiring high energy efficiency, offering a wide range of potential applications.

**Author Contributions:** Conceptualization, W.H.; methodology, W.H. and H.Z.; software, R.W. and Q.C.; validation, W.H. and H.Z.; formal analysis, H.Z.; investigation, W.H. and R.W.; resources, Q.C.; data curation, H.Z.; writing—original draft preparation, W.H. and H.Z.; writing—review and editing, W.H. and Q.C.; visualization, R.W.; supervision, W.H.; project administration, Q.C.; funding acquisition, Q.C. All authors have read and agreed to the published version of the manuscript.

**Funding:** This work was supported by the National Natural Science Foundation of China (Grant No. 62274036), Natural Science Foundation of Fujian Province of China (Grant No. 2022J01079), Science and Technology Plan project of Fujian Province of China (Grant No. 2023H4005).

**Data Availability Statement:** The data that support the findings of this study are available from the corresponding author upon reasonable request.

**Conflicts of Interest:** The authors declare no conflicts of interest.

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
