# Peer review of "A Multi-Bit Quantization Low-Latency Voltage Sense Amplifier Applied in RRAM Computing-in-Memory Macro Circuits"

_electronics, doi:10.3390/electronics13020356_

Round 1

Reviewer 1 Report

Comments and Suggestions for Authors

This work proposes an MQL-VSA that tends to reduce CIM architectures' power consumption and latency. The idea of using a switch-capacitor circuit as a summing point and a 2-bit level quantization showed very apeling results. Nevertheless, some concerns must be addressed first:

1.- There is no transistor sizing methodology for any subcircuit. Are all transistors minimum size? Please, include a Table with these values.

2.- There is no sizing methodology for the capacitors and switches in the sample circuit of Fig. 3. Based on the layout in Fig. 8, the capacitors look to be non-minimum size. A sizing methodology is very important for switch-capacitor circuits due to the operation frequency determines the capacitor and NMOS/PMOS switch sizes. Please, add a set of equations that includes technological parameters like mobility, oxide thickness, vth, etc.

3.- The idea of using a 2-bit quantization method helps to reduce latency and power consumption. However, to implement this strategy, a new circuit that delivers the proper phase timing signals to the sample circuit must be added. Also, these phase timing signals will have a higher frequency than the conventional implementation. Thus, this new circuit will demand more power. Is this power counted in the power consumption report?

4.- The signals shown in Fig. 9 look very clean and well-defined, especially in Q1 and Q1B. Due to switching, I would expect more under or overshoots. Why is that?

5.- Please, run PVT and 200-run Monte Carlo simulation to see the behavior of the proposed circuit. The authors may use the number of successful data/code conversions as a metric, or any other, but deeply explain the reasons for the selected metric.

Reviewer 2 Report

Comments and Suggestions for Authors

The manuscript entitled, "A Multi-bit Quantization Low-latency Voltage-Sense Amplifier Applied in RRAM Computing-in-Memory Macro Circuits", presents a multi-bit quantization low-latency voltage sense amplifier to address high latency and energy consumption in conventional sense amplifiers. The multi-bit quantization is suggested to reduce the number of operational states and optimize read latency in the proposed circuit. Additionally, the manuscript suggests the proposed design reduces the complexity of the sequential control signals and optimizes readout latency performance. As a reviewer, below are my comments related to the presentation and content of the manuscript:

1. The experimental validation of the proposed MQL-VSA circuit was conducted using the 180nm CMOS process with a supply voltage of Vdd = 1.8 V.  Given the current technology trend, the result should be validated using recent technology nodes such as 130nm or lesser. This will be useful in comparing the experimental results achieved in this work with recent works.
2. The proposed design achieved a reduction in chip area as per the manuscript. Although achievement in terms of the area has been discussed, results should be numerically compared in terms of area for the proposed circuit.
3. The experimental results compared in Table 1 demonstrate that the power consumption in the proposed circuit decreased from 90.42 μW in conventional VSA to 70.64 μW in MQL-VSA, however, the power consumption is still significantly higher than [7], [8], and [9].
4. In Section 3, the authors mentioned, "The proposed MQL-VSA, as shown in Figure 2, comprises four main components: ...". In this statement, please correct the figure number as the proposed MQL-VSA circuit is reported in Figure 3 in the manuscript.
5. Please correct the subscripts of the voltages (abbreviations) reported in Section 3, line 108. These subscripts are not clearly reported in line 108.
6. In Section 3, the authors mentioned, "(4) The LSB Detecting circuit, a 2-to-1 selector, selects between OUT2 and OUT2B based on the value ...". Here, based on Figure 3, the LSB Detecting circuit selects an output between OUT2 and not (OUT2B) (as OUT2B is passed through an inverter).
7. In Section 4.2, the performance of MQL-VSA is compared with conventional VSA [10]. It would be great to compare the performance in terms of delay and power with other VSAs available in the literature as well.

Minor comments:
1. Please provide full abbreviations to CIM in the abstract section.
2. Please correct the typo in line 109. The extra space between P0 and (P1) should be removed to avoid confusion.
3. It is recommended to revise the manuscript thoroughly to avoid any typos and grammatical errors. For instance, in Section 3.2, the authors mentioned, "(1) PH1, the voltage sampling phase."

Comments on the Quality of English Language

Revising the manuscript thoroughly is recommended to avoid typos and grammatical errors. For instance, in Section 3.2, the authors mentioned, “(1) PH1, the voltage sampling phase.”

Round 2

Reviewer 1 Report

Comments and Suggestions for Authors

All my comments have been properly addressed. Congratulations.

Reviewer 2 Report

Comments and Suggestions for Authors

Thank you for addressing the reviewer’s comments and incorporating relevant changes in the revised manuscript. The manuscript has been significantly improved and provides relevant technical information to readers interested in this work.

Comments on the Quality of English Language

The manuscript has been significantly improved.